# On the Sample Complexity of Privately Learning Axis-Aligned Rectangles

Menachem Sadigurschi[*]                    Uri Stemmer[†]

## Abstract

We revisit the fundamental problem of learning Axis-Aligned-Rectangles over a finite grid $X^d \subseteq \mathbb{R}^d$ with differential privacy. Existing results show that the sample complexity of this problem is at most $\min\left\{ d \cdot \log|X| \, , \, d^{1.5} \cdot (\log^* |X|)^{1.5} \right\}$. That is, existing constructions either require sample complexity that grows linearly with $\log|X|$, or else it grows super linearly with the dimension $d$. We present a novel algorithm that reduces the sample complexity to only $\widetilde{\mathcal{O}}\left( d \cdot (\log^* |X|)^{1.5} \right)$, attaining a dimensionality optimal dependency without requiring the sample complexity to grow with $\log|X|$. The technique used in order to attain this improvement involves the deletion of "exposed" data-points on the go, in a fashion designed to avoid the cost of the adaptive composition theorems. The core of this technique may be of individual interest, introducing a new method for constructing statistically-efficient private algorithms.

## 1 Introduction

Differential privacy [Dwork et al., 2006] is a mathematical definition for privacy, that aims to enable statistical analyses of databases while providing strong guarantees that individual-level information does not leak. More specifically, consider a database containing data pertaining to individuals, and suppose that we have some data analysis procedure that we would like to apply to this database. We say that this procedure preserves *differential privacy* if no individual's data has a significant effect on the distribution of the outcome of the procedure. Intuitively, this guarantees that whatever is learned about an individual from the outcome of the computation could also be learned with her data arbitrarily modified (or without her data). Formally,

**Definition 1.1** (Dwork et al. [2006]). *A randomized algorithm $\mathcal{A}$ is $(\varepsilon, \delta)$-differentially private if for every two databases $S, S'$ that differ on one row (such databases are called* neighboring*), and every set of outcomes $F$, we have $\Pr[\mathcal{A}(S) \in F] \leq e^\varepsilon \cdot \Pr[\mathcal{A}(S') \in F] + \delta$. The definition is referred to as* pure *differential privacy when $\delta = 0$, and* approximate *differential privacy when $\delta > 0$.*

Over the last decade, we have witnessed an explosion of research on differential privacy, and by now it is largely accepted as a gold-standard for privacy preserving data analysis. In particular, there has been a lot of interest in designing *private learning algorithms*, which are learning algorithms that guarantee differential privacy for their training data. Intuitively, this guarantees that the outcome of the learner (the identified hypothesis) leaks very little information on any particular point from the training set. Works in this vein include [Kasiviswanathan et al., 2011, Beimel et al., 2014, 2019b, 2016, 2020, Bun et al., 2015, Feldman and Xiao, 2015, Bun et al., 2019, Beimel et al., 2019a, Kaplan et al., 2019, 2020a, Alon et al., 2020, Kaplan et al., 2020b, Bun et al., 2020, Alon et al., 2019], and much more.

---

[*]Department of Computer Science, Ben-Gurion University of the Negev. `sadigurs@post.bgu.ac.il`.

[†]Blavatnik School of Computer Science, Tel Aviv University and Google Research. `u@uri.co.il`.

35th Conference on Neural Information Processing Systems (NeurIPS 2021).

However, in spite of the dramatic progress made in recent years on the theory and practice of private learning, much remains unknown and answers to fundamental questions are still missing. In this work, we revisit one such fundamental open question, specifically,

**Question 1.2.** *What is the sample complexity of learning axis-aligned rectangles with privacy?*

Non-privately, learning axis aligned rectangles is one of the most simple and basic of learning tasks, often given as *the first* example for PAC learning in courses or teaching books. Nevertheless, somewhat surprisingly, the sample complexity of learning axis-aligned rectangles with differential privacy is not well-understood. In this work we make a significant progress towards understanding this basic question.

## 1.1 Existing and New Results

Recall that the VC dimension of the class of all axis-aligned rectangles over $\mathbb{R}^d$ is $O(d)$, and hence a sample of size $O(d)$ suffices to learn axis-aligned rectangles non-privately (we omit throughout the introduction the dependency of the sample complexity in the accuracy, confidence, and privacy parameters). In contrast, it turns out that with differential privacy, learning axis-aligned rectangles over $\mathbb{R}^d$ is impossible, even when $d = 1$ [Feldman and Xiao, 2015, Bun et al., 2015, Alon et al., 2019]. In more detail, let $X = \{1, 2, \ldots, |X|\}$ be a finite (one dimensional) grid, and consider the task of learning axis-aligned rectangles over the finite $d$-dimensional grid $X^d \subseteq \mathbb{R}^d$. In other words, consider the task of learning axis-aligned rectangles under the promise that the underlying distribution is supported on (a subset of) the finite grid $X^d$.

For pure-private learning, Feldman and Xiao [2015] showed a lower bound of $\Omega\left(d \cdot \log |X|\right)$ on the sample complexity of this task. This lower bound is tight, as a pure-private learner with sample complexity $\Theta\left(d \cdot \log |X|\right)$ can be obtained using the generic upper bound of Kasiviswanathan et al. [2011]. This should be contrasted with the non-private sample complexity, which is independent of $|X|$.

For approximate-private learning, Beimel et al. [2016] showed that the dependency of the sample complexity in $|X|$ can be significantly reduced. This, however, came at the cost of increasing the dependency in the dimension $d$. Specifically, the private learner of Beimel et al. [2016] has sample complexity $\tilde{O}\left(d^3 \cdot 8^{\log^* |X|}\right)$. We mention that a dependency on $\log^* |X|$ is known to be necessary [Bun et al., 2015, Alon et al., 2019]. Recently, Beimel et al. [2019a] and Kaplan et al. [2020b] studied the related problem of privately learning *halfspaces* over a finite grid $X^d$, and presented algorithms with sample complexity $\tilde{O}\left(d^{2.5} \cdot 8^{\log^* |X|}\right)$. Their algorithms can be used to privately learn axis-aligned rectangles over $X^d$ with sample complexity $\tilde{O}\left(d^{1.5} \cdot 8^{\log^* |X|}\right)$. This can be further improved using the recent results of Kaplan et al. [2020a], and obtain a differentially private algorithm for learning axis-aligned rectangles over $X^d$ with sample complexity $\tilde{O}\left(d^{1.5} \cdot (\log^* |X|)^{1.5}\right)$. We consider this bound to be the baseline for our work, and we will elaborate on it later.

To summarize, our current understanding of the task of privately learning axis-aligned rectangles over $X^d$ gives us two kinds of upper bounds on the sample complexity: Either $d \cdot \log |X|$ or $d^{1.5} \cdot (\log^* |X|)^{1.5}$. That is, current algorithms either require sample complexity that scales with $\log |X|$, or else it scales super linearly in the dimension $d$. This naturally leads to the following question.

**Question 1.3.** *Is there a differentially private algorithm for learning axis-aligned rectangles with sample complexity that scales linearly in $d$ and asymptotically smaller than $\log |X|$?*

We answer this question in the affirmative, and present the following theorem.

**Theorem 1.4** (informal)**.** *There exists a differentially private algorithm for learning axis-aligned rectangles over $X^d$ with sample complexity $\tilde{O}\left(d \cdot (\log^* |X|)^{1.5}\right)$.*

## 1.2 Baseline Construction using Composition

Before we present the technical ideas behind our construction (obtaining sample complexity linear in $d$), we first elaborate on the algorithm obtaining sample complexity $\tilde{O}\left(d^{1.5} \cdot (\log^* |X|)^{1.5}\right)$, which we consider to be the baseline for this work. This baseline algorithm is based on a reduction to (privately) solving the following problem, called the *interior point problem*.

**Definition 1.5** (Bun et al. 2015). *An algorithm $\mathcal{A}$ is said to solve the* Interior Point Problem *for domain $X$ with failure probability $\beta$ and sample complexity $n$, if for every $m \geq n$ and every database $S$ containing $m$ elements from $X$ it holds that:* $\Pr[\min(S) \leq \mathcal{A}(S) \leq \max(S)] \geq 1 - \beta$.

That is, given a database $S$ containing (unlabeled) elements from a (one dimensional) grid $X$, the interior point problem asks for an element of $X$ between the smallest and largest elements in $S$. The baseline we consider for privately learning axis-aligned rectangles is as follows. Suppose that we have a differentially private algorithm $\mathcal{A}$ for the interior point problem over domain $X$ with sample complexity $n$ (let us ignore the failure probability for simplicity). We now use $\mathcal{A}$ to construct the following algorithm $\mathcal{B}$ that takes a database $S$ containing labeled elements from $X^d$. For simplicity, we assume that $S$ contains "enough" positive elements, as otherwise we could simply return the all-zero hypothesis.

1. For every axis $i \in [d]$:

    (a) Project the positive points in $S$ onto the $i$th axis.

    (b) Let $A_i$ and $B_i$ denote the smallest $n$ and the largest $n$ (projected) points, without their labels.

    (c) Let $a_i \leftarrow \mathcal{A}(A_i)$ and $b_i \leftarrow \mathcal{A}(B_i)$.

2. Return the axis-aligned rectangle defined by the intervals $[a_i, b_i]$ at the different axes.

Now, recall that each application of algorithm $\mathcal{A}$ returns an *interior point* of its input points. Hence, for every axis $i$, it holds that the interval $[a_i, b_i]$ contains (the projection) of all but at most $2n$ of the positive examples in the $i$th axis. Therefore, the rectangle returned in Step 2 contains all but at most $2nd$ of the positive points (and it does not contain any of the negative points, because this rectangle is *contained* inside the target rectangle). So algorithm $\mathcal{B}$ errs on at most $2nd$ of its input points.

Assuming that $|S| \gg 2nd$, we therefore get that algorithm $\mathcal{B}$ has small empirical error. As the VC dimension of the class of axis-aligned rectangles is $O(d)$, this means that algorithm $\mathcal{B}$ is a PAC learner for this class with sample complexity $O(nd)$. The issue here is that algorithm $\mathcal{B}$ executes algorithm $\mathcal{A}$ many times (specifically, $2d$ times). Hence, in order to argue that $\mathcal{B}$ is $(\varepsilon, \delta)$-differentially private, standard composition theorems for differential privacy require each execution of algorithm $\mathcal{A}$ to be done with a privacy parameter of $\approx \varepsilon/\sqrt{2d}$. This, in turn, would mean that $n$ (the sample complexity of algorithm $\mathcal{A}$) needs to be at least $\sqrt{2d}$, which means that algorithm $\mathcal{B}$ errs on $2nd \approx d^{1.5}$ input points, which translates to sample complexity of $|S| \gg d^{1.5}$.

The takeaway from this baseline learner is that in order to reduce the sample complexity to be linear in $d$, we want to bypass the costs incurred from composition. That is, we still want to follow the same strategy (apply algorithm $\mathcal{A}$ twice on every axis), but we want to do it without appealing to composition arguments in the privacy analysis. We now briefly survey two intuitive attempts that fail to achieve this, but are useful for the presentation.

**Failed Attempt #1.** As before, let $\mathcal{A}$ denote an algorithm for the interior point problem over domain $X$ with sample complexity $n$. Consider the following modification to algorithm $\mathcal{B}$ (marked in red). As before, algorithm $\mathcal{B}$ takes a database $S$ containing labeled elements from $X^d$, where we assume for simplicity that $S$ contains "enough" positive elements.

1. For every axis $i \in [d]$:

    (a) Project the positive points in $S$ onto the $i$th axis.

    (b) Let $A_i$ and $B_i$ denote the smallest $n$ and the largest $n$ (projected) points, without their labels.

    (c) Let $a_i \leftarrow \mathcal{A}(A_i)$ and $b_i \leftarrow \mathcal{A}(B_i)$.

    (d) Delete from $S$ all points (with their labels) that correspond to $A_i$ and $B_i$.

2. Return the axis-aligned rectangle defined by the intervals $[a_i, b_i]$ at the different axes.

The (incorrect) idea here is that by adding Step 1d we make sure that each datapoint from $S$ is "used only once", and hence we do not need to pay in composition. In other words, the hope is that if every execution of algorithm $\mathcal{A}$ is done with a privacy parameter $\varepsilon$, then the whole construction would satisfy differential privacy with parameter $O(\varepsilon)$.

The failure point of this idea is that by deleting *one* point from the data, we can create a "domino effect" that effects (one by one) many of the sets $A_i, B_i$ throughout the execution. Specifically, consider

two neighboring datasets $S$ and $S' = S \cup \{(x', y')\}$ for some labeled point $(x', y') \in X^d \times \{0, 1\}$. Suppose that during the execution on $S'$ it holds that $x' \in A_1$. So the additional point $x'$ participates "only" in the first iteration of the algorithm, and gets deleted afterwards. However, since the size of the sets $A_i, B_i$ is fixed, during the execution on $S$ (without the point $x'$) it holds that *a different point* $z$ gets included in $A_1$ instead of $x'$, and this point $z$ is then deleted from $S$ (but it is not deleted from $S'$ during the execution on $S'$). Therefore, also during the second iteration we have that $S$ and $S'$ are not identical (they still differ on one point) and this domino effect can continue throughout the execution. That is, a single data point can affect many of the executions of $\mathcal{A}$, and we would still need to pay in composition to argue privacy.

**Failed Attempt #2.** In order to overcome the previous issue, one might try the following variant of algorithm $\mathcal{B}$.

1. For every axis $i \in [d]$:

    (a) Project the positive points in $S$ onto the $i$th axis.

    (b) Let $\text{size}_{A_i} = 2n + \text{Noise}$ and let $\text{size}_{B_i} = 2n + \text{Noise}$.

    (c) Let $A_i$ and $B_i$ denote the smallest $\text{size}_{A_i}$ and the largest $\text{size}_{B_i}$ (projected) points, respectively, without their labels.

    (d) Let $a_i \leftarrow \mathcal{A}(A_i)$ and $b_i \leftarrow \mathcal{A}(B_i)$.

    (e) Delete from $S$ all points (with their labels) that correspond to $A_i$ and $B_i$.

2. Return the axis-aligned rectangle defined by the intersection of the intervals $[a_i, b_i]$ at the different axes.

The idea now is that the noises we add to the sizes of the $A_i$'s and the $B_i$'s would "mask" the domino effect mentioned above. Specifically, the hope is as follows. Consider the execution of (the modified) algorithm $\mathcal{B}$ on $S$ and on $S' = S \cup \{(x', y')\}$, and let $i$ be the first axis such that $x' \in A_i \cup B_i$ during the execution on $S'$. Suppose w.l.o.g. that $x' \in B_i$. Now, the hope is that if during the execution on $S$ we have that the noisy $\text{size}_{B_i}$ is smaller by 1 than its value during the execution on $S'$, then this eliminates the domino effect we mentioned, because we would not need to add another point instead of $x'$. Specifically, during time $i$, the point $x'$ gets deleted from $S'$, and every other point is either deleted from both $S, S'$ or not deleted from any of them. So after time $i$ the two executions continue identically. Thus, the hope is that by correctly "synchronizing" the noises between the two executions (such that only the size of the "correct" set gets modified by 1) we can make sure that only one application of $\mathcal{A}$ is effected (in the last example – only the execution of $\mathcal{A}(B_i)$ is effected), and so we would not need to apply composition arguments.

Although very convincing, this idea fails. The (very subtle) issue here is that it is not clear how to synchronize the noises between the two executions. To see the problem, let us try to formalize the above argument.

Fix two neighboring databases $S$ and $S' = S \cup \{(x', y')\}$. Let us write $A_i, B_i$ and $A'_i, B'_i$ to denote these sets during the executions on $S$ and on $S'$, respectively. Aiming to synchronize the two executions, let us define a mapping $\pi : \mathbb{R}^{2d} \to \mathbb{R}^{2d}$ from noise vectors during the execution on $S'$ to noise vectors during the execution on $S$ (determining the values of $\text{size}_{A_1}, \text{size}_{B_1}, \ldots, \text{size}_{A_d}, \text{size}_{B_d}$), such that throughout the execution we have that $A_i = A'_i$ and $B_i = B'_i$ for all $i$ except for a single pair, say $B_j \neq B'_j$, of neighboring sets.

The straightforward way for defining such a mapping is as follows: Let $j$ be the first time step in which the additional point $x'$ gets included in a set $A'_j$ or $B'_j$, and say that it is included in $B'_j$. Then the mapping would be to reduce (by 1) the value of $\text{size}_{B_j}$ (the noisy size of $B_j$ during the execution on $S$). This would indeed make sure that, conditioned on the noise vectors $v'$ and $v = \pi(v')$, the two executions differ only in a single application of the interior point algorithm $\mathcal{A}$, and hence the outcome distribution of these two (conditioned) executions are very similar (in the sense of differential privacy). That is, for any noise vector $v$ and any event $F$,

$$\Pr[\mathcal{B}(S') \in F | v] \leq e^\varepsilon \cdot \Pr[\mathcal{B}(S) \in F | \pi(v)] + \delta.$$

Furthermore, (assuming an appropriate noise distribution) we can make sure that the probability of obtaining the noise vectors $v$ and $\pi(v)$ are similar, with densities differing by at most an $e^\varepsilon$ factor (as

is standard in the literature of differential privacy). Therefore, *had the mapping $\pi$ we defined was a bijection*, for any event $F$ we would have that

$$
\begin{aligned}
\Pr[\mathcal{B}(S') \in F] &= \sum_v \Pr[v] \cdot \Pr[\mathcal{B}(S') \in F | v] \\
&\leq \sum_v e^\varepsilon \cdot \Pr[\pi(v)] \cdot (e^\varepsilon \cdot \Pr[\mathcal{B}(S) \in F | \pi(v)] + \delta) \\
&= \sum_{\pi(v)} e^\varepsilon \cdot \Pr[\pi(v)] \cdot (e^\varepsilon \cdot \Pr[\mathcal{B}(S) \in F | \pi(v)] + \delta) \\
&= e^{2\varepsilon} \cdot \Pr[\mathcal{B}(S) \in F] + e^\varepsilon \cdot \delta,
\end{aligned}
$$

which would be great. Unfortunately, the mapping $\pi$ we defined is *not* a bijection, and hence the second-to-last equality above is incorrect. To see that it is not a bijection, suppose that $d = 2$ and consider a database $S$ containing the following positively labeled points: Many copies of the point $(0, 0)$, as well as 10 copies of the point $(1, 0)$ and 10 copies of the point $(0, 1)$. The neighboring database $S'$ contains, in addition to all these points, also the point $\left(\frac{1}{2}, \frac{1}{2}\right)$. Now suppose that during the execution on $S'$ we have that $|B'_1| = 5$ and $|B'_2| = 4$. That is, the additional point is included in $B'_1$. During the execution on $S$ we therefore reduce (by 1) the size of $B_1$ and so $|B_1| = |B_2| = 4$. Now suppose that during the execution on $S'$ we have that $|B'_1| = 4$ and $|B'_2| = 5$. Here, during the execution on $S$ we reduce the size of $B_2$ and so, again, $|B_1| = |B_2| = 4$. This shows that the mapping $\pi$ we defined is *not* a bijection. In general, in $d$ dimensions, it is only a $d$-to-1 mapping, which would would break our analysis completely (it will not allow us to avoid the extra factor in $d$).

## 1.3 Our Solution - A Technical Overview

We now present a simplified version of our construction, that overcomes the challenges mentioned above. We stress that the actual construction is a bit different. Consider the following (simplified) algorithm.

1. For every axis $i \in [d]$:

   (a) Project the positive points in $S$ onto the $i$th axis.

   (b) Let $\mathrm{size}_{A_i} = 100n + \mathrm{Noise}$ and let $\mathrm{size}_{B_i} = 100n + \mathrm{Noise}$, where the standard deviation of these noises is, say, $10n$.

   (c) Let $A_i$ and $B_i$ denote the smallest $\mathrm{size}_{A_i}$ and the largest $\mathrm{size}_{B_i}$ (projected) points, respectively, without their labels.

   (d) Let $A_i^{\mathrm{inner}} \subseteq A_i$ be the $n$ *largest* points in $A_i$. Similarly, let $B_i^{\mathrm{inner}} \subseteq B_i$ be the $n$ *smallest* points in $B_i$.

   (e) Let $a_i \leftarrow \mathcal{A}(A_i^{\mathrm{inner}})$ and $b_i \leftarrow \mathcal{A}(B_i^{\mathrm{inner}})$.

   (f) Delete from $S$ all points (with their labels) whose projection onto the $i$th is *not* in the interval $[a_i, b_i]$.

2. Return the axis-aligned rectangle defined by the intersection of the intervals $[a_i, b_i]$ at the different axes.

There are *two* important modifications here. First, we still add noise to the size of the sets $A_i, B_i$, but we only use the $n$ "inner" points from these sets. Second, we delete elements from $S$ not based on them being inside $A_i$ or $B_i$, but only based on the (privately computed) interval $[a_i, b_i]$. We now elaborate on these ideas, and present a (simplified) overview for the privacy analysis. Any informalities made herein are removed in the sections that follow.

Let $S$ and $S' = S \cup \{(x', y')\}$ be neighboring databases, differing on the labeled point $(x', y')$. Consider the execution on $S$ and on $S'$. The privacy analysis is based on the following two lemmas.

**Lemma 1.6** (informal). *With probability at least $1 - \delta$, throughout the execution it holds that $x'$ participates in at most $O(\log(1/\delta))$ sets $A_i, B_i$.*

This lemma holds because of our choice for the noise magnitude. In more detail, given that $x' \in A_i$, there is a constant probability that $x' \in A_i \setminus A_i^{\mathrm{inner}}$. Since the interior point $a_i$ is computed from

$A_i^{\text{inner}}$, in such a case we will have that $x' < a_i$, and hence, $x'$ is deleted from the data during this iteration. This means that every time $x'$ is included in $A_i$, there is a constant probability that $x'$ will be deleted from the data. Thus, one can show (using concentration bounds) that the number of times $i$ such that $x' \in A_i$ is bounded (w.h.p.). A similar argument also holds for $B_i$.

**Lemma 1.7** (informal). *In iterations $i$ in which $x'$ is* not *included in $A_i$ or $B_i$, we have that $a_i$ and $b_i$ are distributed* exactly *the same during the execution on $S$ and on $S'$.*

Indeed, in such an iteration, the point $x'$ has no effect on the outcome distribution of $\mathcal{A}$ (who computes $a_i, b_i$). Overall, w.h.p., there are at most $O(\log \frac{1}{\delta})$ axes the point $x'$ effects. We pay in composition only for those axes, while in all other axes we get privacy "for free". This allows us to save a factor of $\sqrt{d}$ in the sample complexity, and obtain an algorithm with sample complexity linear in $d$.

Note that the definition of privacy we work with is that of $(\varepsilon, \delta)$-differential privacy. In contrast to the case of $(\varepsilon, 0)$-differential privacy, where it suffices to analyze the privacy loss w.r.t. every *single* possible outcome, with $(\varepsilon, \delta)$-differential privacy we must account for arbitrary events. To tackle this, we had to perform a more explicit and meticulous analysis than that outlined above. Our analysis draws its structure from the proof of the advanced-composition theorem [Dwork et al., 2010], but instead of composing everything we aim to preform *effective composition*, meaning that we incurr a privacy loss only on a small fraction of the iterations. To achieve this, as we mentioned, we partition the iterations into several types – iteration on which we "pay" in privacy and iterations on which we do not. However, this partition must be done carefully, as the partition itself is random and needs to be different for different possible outcomes.

We believe that ideas from our work can be used more broadly, and hope that they find new applications in avoiding (or reducing) composition costs in other settings.

**Remark 1.8.** *To simplify the presentation, in the technical sections of this paper we assume that the target rectangle is placed at the origin. Our results easily extend to arbitrary axis-aligned rectangles.*

**Notations.** Two datasets $S, S' \in \mathcal{X}$ are said to be *neighboring* if they differ exactly on one element, formally, $d_H(S, S') = 1$. Given a number $\ell \in \mathbb{N}$ and a dataset $S$ containing points from an ordered domain, we use $\min(S, \ell)$ (or $\max(S, \ell)$) to indicate the subset of $\ell$ minimal (or maximal) values within $S$. When $S$ contains points from a $d$-dimentional domain, we write $\min_i(S, \ell)$ (or $\max_i(S, \ell)$) to denote the subset of $\ell$ minimal (or maximal) values within $S$ w.r.t. the $i^{th}$ axis. We write $\text{Lap}(\mu, b)$ to denote the *Laplase distribution* with mean $\mu$ and scale $b$, when the mean is zero we will simply write $\text{Lap}(b)$. See the full version of this paper for additional preliminaries from learning theory.

## 2 The Algorithm

In this work we investigate the problem of privately learning the class of axis-aligned rectangles, defined as follows.

**Definition 2.1** (Axis Aligned Rectangles). *Let $\mathcal{X} = \{0, \ldots, X\}^d$ be a finite discrete d-dimensional domain. Every $p = (p_1, \ldots, p_d) \in \mathcal{X}$, induces a classifier $h_p : \mathcal{X} \to \{0, 1\}$ s.t for a given input $x \in \mathcal{X}$ we have*

$$h_p(x) = \begin{cases} 1, & \forall i \in [d] : x_i \leq p_i \\ 0, & otherwise \end{cases}$$

*Define the class of all* axis-aligned and origin-placed rectangles *as $REC_d^X = \{h_p : p \in \mathcal{X}\}$.*

Let $\mathcal{A}$ be an $(\varepsilon, \delta)$-differentially private algorithm for solving the interior point problem over domain $\{0, \ldots, X\}$ with failure probability $\beta$ and sample complexity $IP_\mathcal{A}(\varepsilon, \delta, \beta)$. We propose Algorithm 1, which we call `RandMargins`, and prove the following theorem.

**Theorem 2.2.** *Let $\varepsilon < 1, \delta < \frac{1}{e^2}, \alpha, \beta$. Algorithm 1 is $(\alpha, \beta, \tilde{\varepsilon}, \tilde{\delta})$-PPAC learner, for the $REC_d$ class, given a labeled sample of size $\mathcal{O}\left(IP_\mathcal{A}(\varepsilon, \delta, \beta) \cdot \frac{d}{\alpha} \log\left(\frac{1}{\alpha}\right) \log\left(\frac{1}{\beta}\right)\right)$, for $\tilde{\delta} = (d+2)\delta$, and $\tilde{\varepsilon} = \mathcal{O}\left(\varepsilon \log(1/\delta)\right).$*

---

[3]Note that the use of the Laplace noise is different from the standard use in the literature of differential privacy. Unlike the standard use of the Laplace noise, we do not use it in order to obtain a private estimation for some real-valued function of the data. We use it in order to mask the domino effect mentioned in the introduction. Hence, it is not proportional to the sensitivity, but rather to the output of the $\mathcal{A}$ algorithm.

---

**Algorithm 1:** `RandMargins`

---

**Input:** Data $S \subseteq \mathbb{R}^d$ of size $n$, and parameters $\beta < \frac{1}{4}$ and $\delta < 1/e^2, \varepsilon$
**Tool used:** An $(\varepsilon, \delta)$-private algorithm $\mathcal{A}$ for solving the interior point problem with failure probability $\beta$ and sample complexity $IP_{\mathcal{A}}(\varepsilon, \delta, \beta)$.

Denote $\Delta = IP_{\mathcal{A}}(\varepsilon, \delta, \beta)$
Denote $\mu = 4\Delta \log(1/\beta)$
Initialize $\bar{S} \leftarrow \{x \in S \mid \text{x is labeled 1}\}$
**for** $i = 1$ **to** $d$ **do**
    $w_i \sim Lap(2\Delta)$    3
    $B_i = \max_i(\bar{S}, \lceil \mu + w_i \rceil)$
    $D_i = \min_i(B_i, \Delta)$
    $p_i \leftarrow \mathcal{A}(D_i, \varepsilon, \delta, \beta)$
    $R_i = \{y \in \bar{S} : y[i] \geq p_i\}$
    $\bar{S} \leftarrow \bar{S} \setminus R_i$
**end for**
**Return** $(p_1, \ldots, p_d)$

---

**Remark 2.3.** *Kaplan et al. [2020a] introduced an algorithm $\mathcal{A}$ for the interior point problem with sample complexity $IP_{\mathcal{A}}(\varepsilon, \delta, \beta) = \widetilde{\mathcal{O}}\left( \frac{1}{\varepsilon} \log^{1.5}\left(\frac{1}{\delta}\right) \left(\log^*(|X|)\right)^{1.5} \right)$. Hence, using their algorithm within Algorithm 1 provides the result of Theorem 1.4.*

We analyze the privacy guarantees of Algorithm 1 in Section 3, and show the following lemma.

**Lemma 2.4.** *Let $\varepsilon$ and $\delta < \frac{1}{e^2}$, given a labeled sample of size $\mathcal{O}\left( IP_{\mathcal{A}}(\varepsilon, \delta, \beta) \cdot \frac{d}{\alpha} \log\left(\frac{1}{\alpha}\right) \log\left(\frac{1}{\beta}\right) \right)$, Algorithm 1 is $(\tilde{\varepsilon}, \tilde{\delta})$-differentially private, for $\tilde{\delta} = (d+2)\delta$, and $\tilde{\varepsilon} = \mathcal{O}(\varepsilon \log(1/\delta))$.*

We analyze the utility guarantees of Algorithm 1 in Section 4, and show the following lemma.

**Lemma 2.5.** *Let $\alpha, \beta, \varepsilon, \delta$, given a labeled sample of size $\mathcal{O}\left( IP_{\mathcal{A}}(\varepsilon, \delta, \beta) \cdot \frac{d}{\alpha} \log\left(\frac{1}{\alpha}\right) \log\left(\frac{1}{\beta}\right) \right)$ with probability at least $1 - \beta$ Algorithm 1 is $\alpha$-accurate.*

## 3 Privacy Analysis

*Proof of Lemma 2.4.* Let $S$ and $S' = S \cup \{(x', y')\}$ be neighboring databases, differing on the labeled point $(x', y')$. Consider the execution on $S$ and on $S'$.

We denote by $ind_i(x)$ the position of the point $x$ in the remaining data $\bar{S}$, when the data is sorted by the $i^{th}$ coordinate.

Denote by $i^*$ the first iteration on which $x'[i] > p_i$, note that $i^*$ is a random variable. For an input set $S$, denote by $\bar{S}_i$ the remaining set at the beginning of the $i^{th}$ iteration and its size by $\bar{n}$.

Partition the iterations in the following way: $\mathcal{I}_{in} = \{i \leq i^* \mid x' \in B_i'\}$, $\quad \mathcal{I}_{out} = \{i < i^* \mid x' \notin B_i'\}$, $\mathcal{I}_{after} = \{i \mid i > i^*\}$.

We first argue that $|\mathcal{I}_{in}|$ is small (with high probability). Intuitively, this follows from the fact that conditioned on $x' \in B_i'$, with constant probability, we get that $x' \in B_i' \setminus D_i$. Note that in such a case, projecting on the $i^{th}$ axis, $x'$ is bigger (or equal) than any point in $D_i$. Furthermore, as the interior point $p_i$ is computed from $D_i$, w.h.p. we get that $x'[i] \geq p_i$, and hence $x'$ is removed from the data. To summarize, conditioned on $x' \in B_i'$ there is a constant probability that $x'$ is removed from the data, and hence the number of times such that $x' \in B_i'$ must be small (w.h.p.). We make this argument formal in the full version of this paper, obtaining the following claim.

**Claim 3.1.** $\Pr[|\mathcal{I}_{in}| > 35 \log(1/\delta)] \leq \delta$.

Next, we will denote by $\mathcal{B}$ the inner steps of the loop in the algorithm. Meaning, the input is $\bar{S}_i$, which $\mathcal{B}$ uses, along with the random noise and the mechanism $\mathcal{A}$, in order to output $p_i$. We now argue,

briefly, that $\mathcal{B}$ is $(\varepsilon, \delta)$-differentially private for every fixture of $w_i$. This, in particular, would mean that $\mathcal{B}$ is $(\varepsilon, \delta)$-differentially private. So fix $w_i$, and fix two neighboring inputs $\bar{S}_i, \bar{S}_i'$ to algorithm $\mathcal{B}$. Since $\bar{S}_i, \bar{S}_i'$ are neighboring, we also have that $B_i$ and $B_i'$ are neighboring (the sets containing the top $\mu + w_i$ elements in $\bar{S}_i$ and $\bar{S}_i'$, respectively). By the same reasoning, we also have that $D_i$ and $D_i'$ are neighboring. As a result, the outcome distributions of the given mechanism $\mathcal{A}$, on $D_i$ and on $D_i'$ are $(\varepsilon, \delta)$-indistinguishable, showing that $\mathcal{B}$ is $(\varepsilon, \delta)$-differentially private.

For convenience, we will assume that the $\mathcal{B}$'s output includes the noise value $w_i$, and that the final output of `RandMargins` includes the noise vector $w = (w_1, \ldots, w_d)$. As will be proven below, algorithm `RandMargins` remains differentially private even when releasing this noise vector (in addition to the output $(p_1, \ldots, p_d)$).

**Lemma 3.2** (Vadhan [2017]). *For every $(\varepsilon, \delta)$-private algorithm $M$ and every two neighboring datasets $S, S'$, there exist an event $G = G(M, S, S')$ such that (i) $\Pr[M(S) \in G] > 1 - \delta$, and (ii) $\Pr[M(S') \in G] > 1 - \delta$, and (iii) $\forall x \in G : \left| \ln\left( \frac{\Pr(M(S)=x)}{\Pr(M(S')=x)} \right) \right| \leq \varepsilon$.*

Define the event $G = \{(p, w) \mid \forall j \in [d] : (p_j, w_j) \in G(\mathcal{B}, \bar{S}_j, \bar{S}_j')\}$, where $G(\mathcal{B}, \bar{S}_j, \bar{S}_j')$ is the event guaranteed to exist by applying Lemma 3.2 to $\mathcal{B}, \bar{S}_j, \bar{S}_j'$.

Note that by Lemma 3.2 and the union bound $\Pr[G] \geq 1 - d\delta$.

We wish to prove that for any possible output set $P$, it holds that

$$\Pr[\texttt{RandMargins}(S) \in P] \leq e^{\tilde{\varepsilon}} \cdot \Pr[\texttt{RandMargins}(S') \in P] + \tilde{\delta}.$$

Define the set

$$R = \left\{ (p, w) \,\middle|\, \ln\left( \frac{\Pr[\mathcal{RM}(S) = (p, w)]}{\Pr[\mathcal{RM}(S') = (p, w)]} \right) > \tilde{\varepsilon} \right\},$$

where $\mathcal{RM}$ is an abbreviation for `RandMargins`.

Now note that for every event $P$,

$$
\begin{aligned}
\Pr[\mathcal{RM}(S) \in P] &\leq \Pr[\mathcal{RM}(S) \in R] + \Pr[\mathcal{RM}(S) \in P \setminus R] \\
&\leq \Pr[\mathcal{RM}(S) \in R] + e^{\tilde{\varepsilon}} \Pr[\mathcal{RM}(S') \in P \setminus R] \\
&\leq \Pr[\mathcal{RM}(S) \in R] + e^{\tilde{\varepsilon}} \Pr[\mathcal{RM}(S') \in P]
\end{aligned}
$$

So it is down to show that $\Pr[\mathcal{RM}(S) \in R] \leq \tilde{\delta}$. That is, we need to prove that

$$\Pr_{p,w \leftarrow \mathcal{RM}(S)} \left[ \ln\left( \frac{\Pr(\mathcal{RM}(S) = p, w)}{\Pr(\mathcal{RM}(S') = p, w)} \right) > \tilde{\varepsilon} \right] \leq \tilde{\delta}.$$

We calculate,

$$
\Pr_{p,w \leftarrow \mathcal{RM}(S)} \left[ \ln\left( \frac{\Pr(\mathcal{RM}(S) = p, w)}{\Pr(\mathcal{RM}(S') = p, w)} \right) > \tilde{\varepsilon} \right] =
$$
$$
\Pr \left[ \left( \ln\left( \frac{\Pr(\mathcal{RM}(S) = p, w)}{\Pr(\mathcal{RM}(S') = p, w)} \right) \cdot \mathbb{1}_{p,w \in G} > \tilde{\varepsilon} \right) \text{ OR } \left( \ln\left( \frac{\Pr(\mathcal{RM}(S) = p, w)}{\Pr(\mathcal{RM}(S') = p, w)} \right) \cdot \mathbb{1}_{p,w \notin G} > \tilde{\varepsilon} \right) \right]
$$
$$
\leq \Pr_{p,w \leftarrow \mathcal{RM}(S)} \left[ \ln\left( \frac{\Pr(\mathcal{RM}(S) = p, w)}{\Pr(\mathcal{RM}(S') = p, w)} \right) \cdot \mathbb{1}_{p,w \in G} > \tilde{\varepsilon} \right]
$$
$$
+ \Pr_{p,w \leftarrow \mathcal{RM}(S)} \left[ \ln\left( \frac{\Pr(\mathcal{RM}(S) = p, w)}{\Pr(\mathcal{RM}(S') = p, w)} \right) \cdot \mathbb{1}_{p,w \notin G} > \tilde{\varepsilon} \right]
$$
$$
\leq \Pr_{p,w \leftarrow \mathcal{RM}(S)} \left[ \ln\left( \frac{\Pr(\mathcal{RM}(S) = p, w)}{\Pr(\mathcal{RM}(S') = p, w)} \right) \cdot \mathbb{1}_{p,w \in G} > \tilde{\varepsilon} \right] + (1 - \Pr[G])
$$
$$
\leq \Pr_{p,w \leftarrow \mathcal{RM}(S)} \left[ \ln\left( \frac{\Pr(\mathcal{RM}(S) = p, w)}{\Pr(\mathcal{RM}(S') = p, w)} \right) \cdot \mathbb{1}_{p,w \in G} > \tilde{\varepsilon} \right] + d\delta
$$

It remains to prove that $\Pr_{p,w\leftarrow\mathcal{RM}(S)}\left[\ln\left(\frac{\Pr(\mathcal{RM}(S)=p,w)}{\Pr(\mathcal{RM}(S')=p,w)}\right)\cdot\mathbb{1}_{p,w\in G}>\tilde{\varepsilon}\right]\leq\delta$. We calculate,

$$
\Pr_{p,w\leftarrow\mathcal{RM}(S)}\left[\ln\left(\frac{\Pr(\mathcal{RM}(S)=p,w)}{\Pr(\mathcal{RM}(S')=p,w)}\right)\cdot\mathbb{1}_{p\in G}>\tilde{\varepsilon}\right]
$$

$$
=\Pr_{p,w\leftarrow\mathcal{RM}(S)}\left[\ln\left(\prod_{i=1}^{d}\frac{\Pr(\mathcal{RM}(S)_i=p_i,w_i\mid p_{<i},w_{<i})}{\Pr(\mathcal{RM}(S')_i=p_i,w_i\mid p_{<i},w_{<i})}\right)\cdot\mathbb{1}_{p,w\in G}>\tilde{\varepsilon}\right]
$$

$$
=\Pr_{p,w\leftarrow\mathcal{RM}(S)}\left[\sum_{i=1}^{d}\ln\left(\frac{\Pr(\mathcal{RM}(S)_i=p_i,w_i\mid p_{<i},w_{<i})}{\Pr(\mathcal{RM}(S')_i=p_i,w_i\mid=p_{<i},w_{<i})}\right)\cdot\mathbb{1}_{p,w\in G}>\tilde{\varepsilon}\right]
$$

$$
\leq\Pr_{p,w\leftarrow\mathcal{RM}(S)}\left[\sum_{i=1}^{d}\left(\ln\left(\frac{\Pr(\mathcal{RM}(S)_i=p_i,w_i\mid p_{<i},w_{<i})}{\Pr(\mathcal{RM}(S')_i=p_i,w_i\mid p_{<i},w_{<i})}\right)\cdot\mathbb{1}_{p_i,w_i\in G_i(\mathcal{B},\bar{S}_i,\bar{S}'_i)}\right)>\tilde{\varepsilon}\right]
$$

$$
=\Pr_{p,w\leftarrow\mathcal{RM}(S)}\left[\sum_{i\in\mathcal{I}_{in}}\left(\ln\left(\frac{\Pr(\mathcal{RM}(S)_i=p_i,w_i\mid p_{<i},w_{<i})}{\Pr(\mathcal{RM}(S')_i=p_i,w_i\mid p_{<i},w_{<i})}\right)\cdot\mathbb{1}_{p_i,w_i\in G_i(\mathcal{B},\bar{S}_i,\bar{S}'_i)}\right)\right.
$$

$$
+\sum_{i\in\mathcal{I}_{out}}\left(\ln\left(\frac{\Pr(\mathcal{RM}(S)_i=p_i,w_i\mid p_{<i},w_{<i})}{\Pr(\mathcal{RM}(S')_i=p_i,w_i\mid p_{<i},w_{<i})}\right)\cdot\mathbb{1}_{p_i,w_i\in G_i(\mathcal{B},\bar{S}_i,\bar{S}'_i)}\right)
$$

$$
\left.+\sum_{i\in\mathcal{I}_{after}}\left(\ln\left(\frac{\Pr(\mathcal{RM}(S)_i=p_i,w_i\mid p_{<i},w_{<i})}{\Pr(\mathcal{RM}(S')_i=p_i,w_i\mid p_{<i},w_{<i})}\right)\cdot\mathbb{1}_{p_i,w_i\in G_i(\mathcal{B},\bar{S}_i,\bar{S}'_i)}\right)>\tilde{\varepsilon}\right].^{4}
$$

$$(1)$$

We will prove the following

(i) $\Pr\left[\sum_{i\in\mathcal{I}_{after}}\ln\left(\frac{\Pr(\mathcal{RM}(S)_i=p_i,w_i\mid p_{<i},w_{<i})}{\Pr(\mathcal{RM}(S')_i=p_i,w_i\mid p_{<i},w_{<i})}\right)\cdot\mathbb{1}_{p_i,w_i\in G_i(\mathcal{B},\bar{S}_i,\bar{S}'_i)}=0\right]=1$

(ii) $\Pr\left[\sum_{i\in\mathcal{I}_{out}}\ln\left(\frac{\Pr(\mathcal{RM}(S)_i=p_i,w_i\mid p_{<i},w_{<i})}{\Pr(\mathcal{RM}(S')_i=p_i,w_i\mid p_{<i},w_{<i})}\right)\cdot\mathbb{1}_{p_i,w_i\in G_i(\mathcal{B},\bar{S}_i,\bar{S}'_i)}=0\right]=1$

(iii) $\Pr\left[\sum_{i\in\mathcal{I}_{in}}\ln\left(\frac{\Pr(\mathcal{RM}(S)_i=p_i,w_i\mid p_{<i},w_{<i})}{\Pr(\mathcal{RM}(S')_i=p_i,w_i\mid p_{<i},w_{<i})}\right)\cdot\mathbb{1}_{p_i,w_i\in G_i(\mathcal{B},\bar{S}_i,\bar{S}'_i)}\leq\tilde{\varepsilon}\right]\geq1-\delta$

Combining the above three claims implies a bound on (1) and finishes the proof.

*Proof of (i).* After $i^*$, by the algorithm definition, $x'$ gets removed from $S'$. Hence, for every $i>i^*$, conditioning on $\mathcal{RM}(S)_{<i}=p_{<i}$, it holds that $B'_i=B_i$. This implies that, for every $i\in\mathcal{I}_{after}$,

$$
\Pr(\mathcal{RM}(S)_i=p_i,w_i\mid p_{<i},w_{<i})=\Pr(\mathcal{RM}(S')_i=p_i,w_i\mid p_{<i},w_{<i})
$$

which yields

$$
\frac{\Pr(\mathcal{RM}(S)_i=p_i,w_i\mid p_{<i},w_{<i})}{\Pr(\mathcal{RM}(S')_i=p_i,w_i\mid p_{<i},w_{<i})}=1
$$

$$
\Rightarrow\Pr\left[\sum_{i\in\mathcal{I}_{after}}\ln\left(\frac{\Pr(\mathcal{RM}(S)_i=p_i,w_i\mid p_{<i},w_{<i})}{\Pr(\mathcal{RM}(S')_i=p_i,w_i\mid p_{<i},w_{<i})}\right)\cdot\mathbb{1}_{p_i,w_i\in G_i(\mathcal{B},\bar{S}_i,\bar{S}'_i)}=0\right]=1
$$

*Proof of (ii).* Recall that by the definition of $\mathcal{I}_{out}$ for every $i\in\mathcal{I}_{out}$ it holds that $x'\notin B'_i$, and hence, conditioning on the previous outputs, $B'_i=B_i$. We therefore get that the distribution of the $i^{th}$ output is also the same. Formally,

$$
\Pr(\mathcal{RM}(S)_i=p_i,w_i\mid p_{<i},w_{<i})=\Pr(\mathcal{RM}(S')_i=p_i,w_i\mid p_{<i},w_{<i}).
$$

---

[4] Note that the outer probability is over $p$ and $w$. This allows the partition of the iterations into $\mathcal{I}_{in},\mathcal{I}_{out},\mathcal{I}_{after}$ to be well-defined, as this partition depends on $p,w$.

This results in

$$\ln\left(\frac{\Pr(\mathcal{RM}(S)_i = p_i, w_i \mid p_{<i}, w_{<i})}{\Pr(\mathcal{RM}(S')_i = p_i, w_i \mid p_{<i}, w_{<i})}\right) \cdot \mathbb{1}_{p_i, w_i \in G_i(\mathcal{B}, \bar{S}_i, \bar{S}'_i)} = 0$$

$$\Rightarrow \Pr\left[\sum_{i \in \mathcal{I}_{out}} \ln\left(\frac{\Pr(\mathcal{RM}(S)_i = p_i, w_i \mid p_{<i}, w_{<i})}{\Pr(\mathcal{RM}(S')_i = p_i, w_i \mid p_{<i}, w_{<i})}\right) \cdot \mathbb{1}_{p_i, w_i \in G_i(\mathcal{B}, \bar{S}_i, \bar{S}'_i)} = 0\right] = 1.$$

*Proof of (iii).* Note that, as we assume that the output of $\mathcal{RM}$ includes the random Laplasian noise, then by fixing the past output-point $p_{<i}, w_{<i}$ we also fix $\bar{S}_i, \bar{S}'_i$. So,

$$\ln\left(\frac{\Pr(\mathcal{RM}(S)_i = p_i, w_i \mid p_{<i}, w_{<i})}{\Pr(\mathcal{RM}(S')_i = p_i, w_i \mid p_{<i}, w_{<i})}\right) \cdot \mathbb{1}_{p_i, w_i \in G_i(\mathcal{B}, \bar{S}_i, \bar{S}'_i)}$$

$$= \ln\left(\frac{\Pr(\mathcal{B}(\bar{S}_i) = p_i, w_i)}{\Pr(\mathcal{B}(\bar{S}'_i) = p_i, w_i)}\right) \cdot \mathbb{1}_{p_i, w_i \in G_i(\mathcal{B}, \bar{S}_i, \bar{S}'_i)}.$$

Moreover, by the definition of the events $G_i$ it holds that

$$\Pr\left[\left|\ln\left(\frac{\Pr(\mathcal{B}(\bar{S}_i) = p_i, w_i)}{\Pr(\mathcal{B}(\bar{S}'_i) = p_i, w_i)}\right) \cdot \mathbb{1}_{p_i, w_i \in G_i(\mathcal{B}, \bar{S}_i, \bar{S}'_i)}\right| \leq 2\varepsilon\right] = 1.$$

which yields

$$\Pr\left[\sum_{i \in \mathcal{I}_{in}} \ln\left(\frac{\Pr(\mathcal{B}(\bar{S}_i) = p_i, w_i)}{\Pr(\mathcal{B}(\bar{S}'_i) = p_i, w_i)}\right) \cdot \mathbb{1}_{p_i, w_i \in G_i(\mathcal{B}, \bar{S}_i, \bar{S}'_i)} > \tilde{\varepsilon}\right]$$

$$\leq \Pr\left[\sum_{i \in \mathcal{I}_{in}} 2\varepsilon > \tilde{\varepsilon}\right] \leq \Pr\left[|\mathcal{I}_{in}| > \frac{\tilde{\varepsilon}}{2\varepsilon}\right] \leq \delta,$$

where the last inequality follows from Claim 3.1 and from our choice of $\tilde{\varepsilon} = \mathcal{O}\left(\varepsilon \log(1/\delta)\right)$.

$\square$

## 4  Utility

*Proof of Lemma 2.5.* First, we must ensure that at every iteration, with high probability, we have enough points left in $\bar{S}$. At the same time we must ensure that the axillary algorithm $\mathcal{A}$ will output an inner point of the given subset. Denote $a_j = w_j + \mu$. By the definition of the noise $w$ and the mean $\mu$, we get that for every iteration $i$:   $\Pr[a_i > 6\Delta \log(1/\beta)] < \beta$. Hence, with probability $\geq 1 - d\beta$, it holds that for every $i$ $a_i \leq 6\Delta \log(1/\beta)$. This means that the total number of removed point is at most $6d\Delta \log(1/\beta)$. Therefore, for a sample of size $6d\Delta \log(1/\beta)$ with high probability $\bar{S}$ will contain enough points.

Regarding the algorithm's accuracy, we notice that at every iteration $j$, $\mathcal{A}$ outputs a point which is at least the $a_j$-th largest point from *the points left in the set*. This means that, in the worst case, we delete $a_j$ points from the data set at this iteration. Hence, again in worst case, we will output the $\sum_{j=1}^{i} a_j$-th largest point in the $j^{th}$ axis.

By the above reasoning, with high probability we can say that for every $i$ it holds that $a_j \leq 6\Delta \log(1/\beta)$. Meaning that every $p_j$ is at least the $\sum_{j=1}^{d} a_j \leq 6d\Delta \log(1/\beta)$ largest point in the axis. This implies that, for sample of size $\mathcal{O}\left(\frac{d\Delta}{\alpha} \log(1/\alpha) \log(1/\beta)\right)$, denoting the by $h_p$ the hypothesis induces by the output of Algorithm 1 $\Pr_{S \sim \mathcal{P}^n}[\mathrm{err}_S(h_p) \geq \alpha/2] \leq \beta/2$. Since the VC-dimension of the class $REC_d$ is $2d$, by the VC-theory bounds and the fact that the sample size is at least as the sample complexity bound $\mathcal{O}\left(\frac{1}{\alpha}\left(d\log\left(\frac{1}{\alpha}\right) + \log\left(\frac{1}{\beta}\right)\right)\right)$ it holds that:   $\Pr_{S \sim \mathcal{P}^n}[\mathrm{err}_{\mathcal{P}}(h_p) \geq \mathrm{err}_S(h_p) + \alpha/2] \leq \beta/2$. Combining the two bounds concludes the proof.  $\square$

## Acknowledgments

This research was partially supported by the Israel Science Foundation (grant 1871/19), by Len Blavatnik and the Blavatnik Family foundation, and by the Cyber Security Research Center at Ben-Gurion University of the Negev.

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
