We use standard definitions from statistical learning theory. See, e.g., Shalev-Shwartz and Ben-David [2014]. A classifier is a function $f : \mathcal{X} \to \{0, 1\}$.

**Definition 2.1** (Generalization error). *The generalization error of a classifier $f$ w.r.t. a distribution $\mathcal{P}$ is defined as $\mathrm{err}_{\mathcal{P}}(f) = \Pr_{(x,y) \sim \mathcal{P}}[f(x) \neq y]$.*

We focus on the *realizable* setting in which for a class $\mathcal{H}$ of potential classifiers, there exist some $h^* \in \mathcal{H}$, s.t $\mathrm{err}_{\mathcal{P}}(h^*) = 0$.

**Definition 2.2** (Sample error). *The empirical error of a classifier $f$ w.r.t. a labeled-sample $S \in (\mathcal{X} \times \{0, 1\})^n$ is defined as $\mathrm{err}_S(f) = \frac{1}{n} \sum_{(x,y) \in S} \mathbb{1}[f(x) \neq y]$.*

**Definition 2.3** (PAC learnability Valiant [1984]). *Let $\alpha, \beta \in [0, 1]$ and let $m \in \mathbb{N}$. An algorithm $\mathcal{A}$ is an $(\alpha, \beta, m)$-PAC-learning algorithm for a class $\mathcal{H}$ if for every distribution $\mathcal{P}$ over $\mathcal{X} \times \{0, 1\}$ s.t. $\exists h^* \in \mathcal{H}$ with $\mathrm{err}_{\mathcal{P}}(h^*) = 0$, it holds that $\Pr_{S \sim \mathcal{P}^m}[\mathrm{err}_{\mathcal{P}}(\mathcal{A}(S)) > \alpha] < \beta$. We refer to $m$ as the the sample complexity of $\mathcal{A}$.*

**Definition 2.4** (Private-PAC learnability). *An algorithm $\mathcal{A}$ is an $(\alpha, \beta, \varepsilon, \delta, m)$-PPAC learner for a class $\mathcal{H}$ if: (i) $\mathcal{A}$ is $(\varepsilon, \delta)$-differentially private; and, (ii) $\mathcal{A}$ is an $(\alpha, \beta, m)$-PAC learning algorithm for $\mathcal{H}$.*

**Definition 2.5** (Shattering). *Let $\mathcal{H}$ be a class of functions over a domain $\mathcal{X}$. A set $S = (s_1, \ldots, s_k) \subseteq \mathcal{X}$ is said to be shattered by $\mathcal{H}$ if $|\{(f(s_1), \ldots, f(s_k)) : f \in \mathcal{H}\}| = 2^k$.*

**Definition 2.6** (VC Dimension Vapnik and Chervonenkis [1971]). *The* VC dimension *of a class* $\mathcal{H}$, *denoted as* $VC(\mathcal{H})$, *is the cardinality of the largest set shattered by* $\mathcal{H}$. *If* $\mathcal{H}$ *shatteres sets of arbitrary large cardinality then it is said that* $VC(\mathcal{H}) = \infty$.

**Theorem 2.7** (VC Dimension Generalization Bound Vapnik and Chervonenkis [1971], Blumer et al. [1989]). *Let* $\mathcal{H}$ *be a function-class and let* $\mathcal{P}$ *be a probability measure over* $\mathcal{X} \times \{0,1\}$. *For every* $\alpha, \beta > 0$, *every* $n \in \mathcal{O}\left(\frac{1}{\alpha}\left(VC(\mathcal{H})\log(\frac{1}{\alpha}) + \log(\frac{1}{\beta})\right)\right)$ *and every* $f \in \mathcal{H}$ *it holds that* $\Pr_{S \sim \mathcal{P}^n}[\exists f \in \mathcal{H} : \mathrm{err}_{\mathcal{P}}(f) \geq \alpha \wedge \mathrm{err}_S(f) \leq \alpha/10] \leq \beta$.

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

# A  Proof of Claim 4.1

In order to provide a concentration bound for adaptive cases such as the one at hand, Gupta et al. [2010], described the following "game". We will use a slight variation of their results, stated in [Kaplan et al., 2021].

---

**A $m$ round game**

In each round $i$:

1. The adversary chooses $0 \leq q_i \leq 1/2$ and $q_i/4 \leq \bar{q}_i \leq 1 - q_i$, possibly based on the first $(i-1)$ rounds
2. A random variable $X_i \in \{0, 1, 2\}$ is sampled (and the outcome is given to the adversary), where $\Pr[X_i = 1] = q_i$ and $\Pr[X_i = 1] = \bar{q}_i$ and $\Pr[X_i = 0] = 1 - \bar{q}_i - q_i$

---

Upon that they define the following random variable $Z_i = \mathbb{1}_{\forall j \leq i : X_j \neq 2}$. Intuitively $Z_i$ indicates the status of the adversary, it is 1 from the start up until the adversary "fails". The adversary's goal is to maximize the amount of time-steps on which $X_i = 1$ but his "score" is counted only until the first round when $X_i = 2$.

We will use the following lemma.

**Lemma A.1** ([Gupta et al., 2010, Kaplan et al., 2021]). *For every adversary's strategy,*

$$\Pr\left[\sum_{i=1}^{m} Z_i \mathbb{1}_{X_i=1} > \gamma\right] \leq e^{(-\gamma/5+6)}$$

Denote $q_i = \Pr_{w_i, p_i}[x' \in S_i' \wedge x'[i] < p_i]$ and $\bar{q}_i = \Pr_{w_i, p_i}[x'[i] \geq p_i]$. Let $X_1, \ldots, X_d$ be a series of random variables with $\Pr[X_i = 0] = 1 - q_i - \bar{q}_i$, $\Pr[X_i = 1] = q_i$ and $\Pr[X_i = 2] = \bar{q}_i$ Our goal is to bound the number of steps on which $X_i = 1$. By Lemma A.1, it is indeed bounded, with high probability, as long as the following conditions hold

1. $q_i \leq \frac{1}{2}$

2. $\frac{q_i}{4} \leq \bar{q}_i$.

We shall now prove that the two conditions do hold.

$$
\begin{aligned}
\bar{q}_i &= \Pr_{p_i}[x'[i] > p_i] \\
&\geq \Pr_{w_i, p_{\leq i}}[x'[i] > p_i \mid ind_i(x') \geq \bar{n} - (\mu + w_i)] \cdot \Pr[ind_i(x') \geq \bar{n} - (\mu + w_i)] \\
&\geq \Pr_{w_i, p_{\leq i}}[x'[i] > p_i \mid ind_i(x') \geq \bar{n} - (\mu + w_i)] \cdot q_i \\
&\geq \left( \Pr_{w_i, p_{\leq i}}[x' \in S_i \setminus D_i \mid ind_i(x') \geq \bar{n} - (\mu + w_i)] - \beta \right) \cdot q_i \\
&= \left( \Pr_{w_i, p_{\leq i}}[ind_i(x') \geq \bar{n} - (\mu + w_i) + \Delta \mid ind_i(x') \geq \bar{n} - (\mu + w_i)] - \beta \right) \cdot q_i \\
&\geq \left( \frac{1}{2} - \beta \right) \cdot q_i \qquad (2) \\
&\geq \frac{1}{4} \cdot q_i
\end{aligned}
$$

where (2) holds since $w_i \sim Lap(2\Delta)$. The last inequality is due to the upper bound on $\beta$. For the first condition

$$
\begin{aligned}
q_i &= \Pr_{w_i, p_{\leq i}}[ind_i(x') \geq \bar{n} - (\mu + w_i) \wedge x'[i] < p_i] \\
&\leq \Pr_{w_i, p_{\leq i}}[ind_i(x') \geq \bar{n} - (\mu + w_i) \wedge x' \notin S_i \setminus D_i] + \delta \\
&= \Pr_{w_i, p_{\leq i}}[ind_i(x') \geq \bar{n} - (\mu + w_i) \wedge ind_i(x') < \bar{n} - (\mu + w_i) + \Delta] + \delta \\
&\leq \frac{1}{4} + \delta \\
&\leq \frac{1}{2},
\end{aligned}
$$

when the penultimate inequality holds, as before, by the distribution $w_i$. By Lemma A.1 this proves that
$$ \Pr[|\mathcal{I}_{in}| > \gamma] \leq \exp\left(-\gamma/5 + 6\right). $$
Setting $\gamma = 35 \log(1/\delta)$ we get that
$$ \Pr[|\mathcal{I}_{in}| > 35 \log(1/\delta)] \leq \delta. $$