# OpenReview forum: "On the Sample Complexity of Privately Learning Axis-Aligned Rectangles"
_NeurIPS.cc/2021/Conference — NeurIPS 2021 Poster_

### Official Review · Reviewer_Mzck · 2021-07-08

**Rating:** 6
**Confidence:** 3

**Summary:**

This paper studies the problem of privately learning an axis-aligned rectangle. In private learning, the goal is to approximate an unknown target function while leaking little information about individual training samples. More formally, the learning algorithm is required to be differentially private in the sense that for any two training sets differing in a single training sample, the distributions of the output target functions on the two data sets are (eps,delta)-close in the standard differentially private notion.

Previous work on this problem includes a learning algorithm with sample complexity O(d^1.5 (log* X)^1.5) and one with O(d log X) when the rectangles comes from a d-dimensional grid X^d. Here the dependency on eps, delta and the accuracy of the learned hypothesis has been hidden from the bound, but is polynomial in eps and the accuracy and polylogarithmic in delta.

It is known from a lower bound side that the sample complexity must depend on log*X and grow linearly with d. The goal of this submission is to derive a learning algorithm with samples depending linearly on d and as o(log X). This is achieved by presenting an algorithm needing O(d (log*X)^1.5) samples, thus shaving a sqrt(d) factor (or a logX/log*X factor) from previous work.

The main idea in the algorithm is to use a subroutine for the Interior Point Problem to find a point close to the edges of the rectangle learned. Formally, the IPP asks to output a point lying between the minimum and maximum point in a one-dimensional data base with domain X. This can be done privately with O((log*X)^1.5) samples, ignoring the dependency on eps and delta. The basic idea is to pick the outermost 100n+noise points that lie inside the rectangle along each axis. From these, the n points closest to the interior of the rectangle are selected and the IPP algorithm is run on them to output a point with i'th coordinate inside the rectangle and close to the border.

**Limitations And Societal Impact:**

Yes

**Main Review:**

The paper is well-written, with a clear walk-through of the main ideas of the algorithm and an explanation of the pitfalls one has to avoid. The problem of learning a rectangle is pretty fundamental and the combination with privacy seems well-motivated. The algorithm is a quite simple reduction to IPP, with the tricky part being the analysis.

Improving the sample complexity by a sqrt(d) factor is a nice theoretical contribution, although the many hidden factors depending on eps, delta etc. make the algorithm rather infeasible in practice. At least one has to care about the exact sample complexity of privately learning rectangles to enjoy the paper, as otherwise it feels a bit like parameter-pushing. All in all, I feel the paper helps understand the sample complexity of a fundamental problem, but it is not clear to me that the result will otherwise have much impact. As such, I find that the paper can be accepted at NeurIPS, but is not a clear accept.


**Time Spent Reviewing:**

3

---

> ### Author Response · Authors · 2021-08-10
> **Rebuttal**
>
> Thank you very much for your time and your helpful comments and suggestions. We will take all comments to our attention and make corrections and improvements accordingly. In the following, we respond to a specific point you raised.
>
> "Improving the sample complexity by a sqrt(d) factor is a nice theoretical contribution, although the many hidden factors depending on eps, delta etc. make the algorithm rather infeasible in practice. At least one has to care about the exact sample complexity of privately learning rectangles to enjoy the paper, as otherwise it feels a bit like parameter-pushing."
>
> We agree that this is a theory paper. We believe that our results are significant because of the following three factors:
> (1) Axis-aligned rectangles is one of the most basic of learning tasks, to the extent that it is usually given as *the first* example for PAC learning in courses and in teaching books. Hence, given the recent interest in private learning, we believe that understanding this problem with differential privacy is a must.
> (2) Our quantitative contributions are significant, as we improve the sample complexity by a factor of sqrt(d).
> (3) In the process of pushing bounds forward for this problem, we were forced to develop new algorithmic ideas in order to avoid the costs incurred by composition theorems for differential privacy. Our ideas may find additional applications in future work.

---

### Official Review · Reviewer_wYvK · 2021-07-16

**Rating:** 7
**Confidence:** 4

**Summary:**

The authors investigate the problem of differentially-privately learning Axis-Aligned Rectangles over a finite grid X of dimension d. They demonstrate the existence of a differentially private algorithm for learning axis-aligned rectangles with sample complexity that scales linearly in d and is asymptotically smaller than log |X|. The sample complexity bounds achieved by authors for this problem are better than existing DP works.


**Limitations And Societal Impact:**

Yes

**Main Review:**

Given a sample S containing labeled points from a d-dimensional finite space X, a naive DP algorithm suggested by the authors projects positive points in S onto each axis i (i=1,...,d) and makes two black-box calls (per dimension) to the interior point algorithm of Kaplan,Ligett, Mansour, Naor, Stemmer [COLT’20] with sample complexity of \tilde{O}((log^*|X|)^{1.5}), where the first call is made with the smallest n projected points outputting a_i and the second call is made with the largest n projected points outputting b_i, and the algorithm returns the rectangle defined by the interval [a_i,b_i]. Using composition theorems to analyse this algorithm would give a sample complexity of \tilde{O}(d^{1.5}*(log^*|X|)^{1.5}). The authors reduce this sample complexity to \tilde{O}(d*(log^*|X|)^{1.5}) by adding noise and deleting encountered points appropriately that helps bypass the costs incurred from composition.

In more details, given two neighboring databases S and S’ which differ on a point (x’,y’), the (\eps,\delta)-privacy analysis of the main algorithm only pays a privacy loss for those axes which the differing point x’ directly affects and this is bounded by O(log (1/\delta)), for the rest of the axes, the output distributions of S and S’ are the same. The authors essentially refine the advanced composition theorem to only compose privacy over those iterations for which the differing point is actually present, and thus incur a privacy loss on a small fraction of interactions. This technique is a good technical contribution that can potentially be used in the analysis of other DP algorithms.

It would help to give more clarity on the actual amount of noise added at each step of the main algorithm. For example,  Laplace noise is usually added in proportion to the sensitivity of the function being computed, it is not clear why Laplace noise is added in proportion to the sample complexity of the IP algorithm to \mu.

Clarity

It would help to explain a bit the proof strategy and analysis tools -- this gives  the PC members taking the decision more insight onto the novelty of the paper and elements of originality and difficulty.

Section 1 is written very well and provides a good high-level description of the methods, but Sections 2, 3, 4 and 5 contain many typos and can be more clear.

In Section 2, Differential Privacy should be explained with respect to adding Laplace noise, since this is what is applied in the main algorithm.

In Section 3, the main theorem (Thm 3.2) has undefined variables, the Sample size includes a \Delta parameter that has not been defined in the statement. I assume this is the sample complexity of the IP algorithm, but this should be made clear.
Some typos:
Line 250: typo, should be ``Laplace’’ distribution
Line 295: typo, should be ``differentially’’-private in Lemma 3.4
Line 333: typo, should be ``auxiliary’’ algorithm A


In Sections 4 and 5, the actual proofs for privacy and accuracy can be more clear. For eg., lines 334 and 335, it would be helpful to expand on how the Pr[a_i>_]<\beta is obtained.

Significance

Given the recent results on learning and privacy, this paper will be of interest to the privacy community.




**Time Spent Reviewing:**

2

---

> ### Author Response · Authors · 2021-08-10
> **Rebuttal**
>
> Thank you very much for your time and your helpful comments and suggestions. We will take all comments to our attention and make corrections and improvements accordingly. In the following, we respond to specific points.
>
> --------
>
> "It would help to give more clarity on the actual amount of noise added at each step of the main algorithm"
>
> Agreed. We will elaborate on this point in the next revision.
>
> --------
>
> "Laplace noise is usually added in proportion to the sensitivity of the function being computed, it is not clear why Laplace noise is added in proportion to the sample complexity of the IP algorithm"
>
> You are right that our use of the Laplace noise is very different from the standard way people use it in the literature of differential privacy. Unlike the standard use of the Laplace noise, we do not use it in order to obtain a private estimation for some real-valued function of the data. We use it in order to mask the domino effect mentioned in the introduction. Hence it is not proportional to the sensitivity, but rather to the output of the IP algorithm. We will clarify and elaborate on this point in the next revision.
>
> --------
>
> "It would help to explain a bit the proof strategy and analysis tools"
>
> We made an effort to explain the challenges and the algorithmic ideas in the introduction. We will try to improve this exposition in order to better convey the proof strategy.
>
> --------
>
> "Section 1 is written very well and provides a good high-level description of the methods, but Sections 2, 3, 4 and 5 contain many typos and can be more clear."
>
> Thank you for the many helpful suggestions and typos you spotted. We will fix all of them in the next revision of our paper.

---

> > ### Comment · Reviewer_wYvK · 2021-08-17
> > **response**
> >
> > I am happy with the way the authors addressed my comments and would like to keep my score.

---

### Official Review · Reviewer_qdue · 2021-07-16

**Rating:** 6
**Confidence:** 4

**Summary:**

This paper studies the problem of PAC learning an axis-aligned rectangle in d-dimensions under the constraint of differential privacy (DP). The class of d-dimensional axis-aligned rectangles can be viewed as a generalization of threshold functions. Previous work have shown that the problem of privately learning thresholds is equivalent (up to constants) to the problem of privately selecting the interior point of a set of totally ordered points. A naive approach to extend this result to high-dimensional rectangles is to compute the threshold in each dimension separately and then use advanced composition. Unfortunately the dependence of the samples complexity of this approach on the dimension is $d^{1.5}$ (rather than $d$). Therefore, this problem captures one of the main technical challenges of designing DP methods, namely handling high-dimensional data.

The authors introduce a carefully constructed algorithm that applies the private interior point algorithm on each dimension, but at each step prunes some points in the dataset so that they are not used in future steps. Intuitively, this approach allows them to overcome the expensive advanced composition since not too many points will be reused. [and as a consequence, the resulting method has only linear dependence on $d$]

**Main Review:**

This is an important and fundamental result in private PAC learning. The proof technique that the authors develop is interesting and will be of interest to the general DP community. The authors do a good job in explaining why extended existing solutions for the 1-dimensional case to d-dimensions introduces looseness in the sample complexity bounds. The introduction of the paper is well written. Beyond the introduction, the writing seems to be a bit rushed and that there is a lot of room for improvement in the presentation. Some places where the paper can use some improvement/clarification are highlighted in below (SUBMISSION refers to the line number in the main file, while SUPPLEMENT refers to the line number in the supplementary material)


Line 260-262 (SUBMISSION): In definition 2.3, I believe it would be helpful to explicitly say that “it holds for any $n \geq m$ that:” instead of “it holds that:” since the variable $n$ used in the probability statement at the end of the definition has not been introduced.

Line 286-289 (SUBMISSION): It will be helpful to explain what $\Delta$ is in the statement. The authors can perhaps modify the statement to say something like “Let IP be an algorithm for solving the interior point problem with sample complexity $\Delta$.” Also, in this theorem and in many other Lemma statements, the bounds on some parameters like $\alpha,\beta, \varepsilon$ etc are not explicitly stated.

Comments regarding proof of privacy (Lemma 3.4):

Line 310-313 (SUBMISSION): The authors claim that the first part of the algorithm in the inner loop right before the interior point algorithm is used satisfies $(\varepsilon,0)$-DP, and claim that the inner loop “can be seen as a composition of a Laplace mechanism (represented by the addition of Laplace noise) and the given mechanism $\mathcal{A}$.” It is not immediately clear why this can be viewed as a Laplace mechanism and furthermore, it is not clear how the choice of the scale parameter for the Laplace noise guarantees $(\varepsilon,0)$-DP. The authors should clarify this, as it is key to the correctness of the proof.

Line 359-361 (SUPPLEMENT): Going from the inequality on line 359 to the inequality on line 361 is not immediately obvious to me, even after plugging in the value for z and trying to get the same upper bound as in the equation on line 361.

Line 333 (SUPPLEMENT): (minor question) the authors prove the bound $Pr\left[\ln\left(\frac{\mathcal{RM}(S)=p}{\mathcal{RM}(S’)=p}\right)\cdot \mathbb{I}_{p\in G} \geq \widetilde{\varepsilon}\right] \leq 2\delta$, however I think the upper bound can be $\delta$ without the factor of 2?

Comments regarding proof of accuracy (Lemma 3.5):

1. This proof is relatively straightforward to follow, however I have a few concerns. I don’t see anywhere (neither in the proofs nor in the pseudocode) that explicitly states that the inputs to the algorithm are the positive points in the dataset. I assumed that this is the case since this is what makes the proof make sense to me and this was mentioned very briefly (and informally) in the introduction where the authors stated on Line 91-92 (SUBMISSION): “For simplicity, we assume that $S$ contains “enough” positive elements, as otherwise we could simply return the all-zero hypothesis.”. This statement was not made formal in the paper and I think it is important to explain why it is OK to make this assumption.
2. When analyzing the probability that the empirical error is large, the authors used a union bound on each dimension. So, shouldn’t the term $6d\Delta\log(1/\beta)\log(1/\delta)$ be $6d\Delta\log(2d/\beta)\log(1/\delta)$ to ensure the empirical error is low with probability at least $1-\beta/2$ ?
3. Why did the authors choose $\mu = 4\Delta\log^2(1/\delta)$ instead of $\mu = 4\Delta\log(1/\delta)\log(1/\beta)$?

**Time Spent Reviewing:**

8

---

> ### Author Response · Authors · 2021-08-10
> **Rebuttal**
>
> Thank you very much for your time and your helpful comments and suggestions. We will take all comments to our attention and make corrections and improvements accordingly. In the following, we respond to specific points.
>
> --------
>
> "Beyond the introduction, the writing seems to be a bit rushed and that there is a lot of room for improvement in the presentation."
>
> Thank you for pointing this out. We will add more explanations where it might help improve the presentation.
>
> --------
>
> "Line 260-262: Explicitly say that it holds for any n≥m"
>
> Agreed.
>
> --------
>
> "Line 286-289: Explain what $\Delta$ is in the statement. Also the bounds on some parameters are not explicitly stated."
>
> Agreed. We will fix this in the next revision.
>
> --------
>
> "Line 310-313: The authors claim that the inner loop can be seen as a composition of a Laplace mechanism and the given mechanism A. The authors should clarify this"
>
> We agree that this statement should be better explained, and we will add such an explanation to the next revision of our paper. We include a brief explanation here.  Using the paper’s notation, let B denote the inner steps of the loop of the algorithm (the input of B is $\hat{S}_i$ and its output is $p_i$). We now argue that B is $(\varepsilon,\delta)$-differentially private for every fixture of $w_i$. This, in particular, would mean that B is $(\varepsilon,\delta)$-differentially private. So fix $w_i$, and fix two neighboring inputs S,S’ to algorithm B. Since S,S’ are neighboring, we also have that $S_i$ and $S'_i$ are neighboring (the sets containing the top $\mu+w_i$ elements in $S$ and $S'$, respectively). By the same reasoning, we also have that $D$ and $D'$ are neighboring. As a result, the outcome distributions of the given mechanism A on $D$ and on $D'$ are $(\varepsilon,\delta)$-indistinguishable, showing that B is $(\varepsilon,\delta)$-differentially private.
>
> --------
>
> "Going from the inequality on line 359 to the inequality on line 361 is not immediately obvious to me"
>
> This implication involves a bit of algebraic manipulation. We shall add an intermediate step in order to make it easier to follow.
>
> --------
>
> "Line 333 (SUPPLEMENT): I think the upper bound can be $\delta$ without the factor of 2?"
>
> You are right. The factor 2 is indeed redundant.
>
> --------
>
> "Explicitly state that the inputs to the algorithm are the positive points in the dataset, and explain why it is OK to make this assumption"
>
> You are right. We will make this assumption explicit and justify it in the body of the paper.
>
> --------
>
> "Shouldn’t the term $6d\Delta\log(1/\beta)\log(1/\delta)$ be $6d\Delta\log(2d/\beta)\log(1/\delta)$"
>
> You are correct. We will fix this in the next revision.
>
> --------
>
> "Why did the authors choose $\mu=4\Delta\log^{2}(1/\delta)$ instead of $\mu=4\Delta\log(1/\delta)\log(1/\beta)$?"
>
> This is indeed unnecessary. We will fix this in the next revision.

---

### Official Review · Reviewer_GhmR · 2021-07-20

**Rating:** 9
**Confidence:** 3

**Summary:**

This paper considers the problem of learning axis-aligned rectangles in a differentially private way. It presents a simple and new technique to reduce the sample complexity down to $\tilde{\mathcal{O}}\left(d \cdot (\log^* |X|)^{1.5} \right)$, in contrast to the original result which either has a super-linear dependence in $d$ or grows linearly with $\log |X|$.

**Limitations And Societal Impact:**

No suggestions.

**Main Review:**

This paper is clearly written and technically sound. The proposed solution follows the baseline works previously established by Kaplan et al. , that for each axis, it reduces to an one-dimensional Interior Point Problem. As samples could be repeatedly used for different axis, standard composition theorem from differential privacy would requires $\Omega(d^{1.5})$ many samples. To overcome this issue, this paper tweaks the original reduction by making sure that each sample only participates in one iteration and removing the potential "domino effect", and performs a new theoretical analysis, the efficient composition.

**Time Spent Reviewing:**

5

---

> ### Author Response · Authors · 2021-08-10
> **Rebutel**
>
> Thank you very much for the review and the kind words. We greatly appreciate the time you have devoted to reading and reviewing our paper.

---

### Decision · Program_Chairs · 2021-09-27

**Decision:**

Accept (Poster)

**Comment:**

This paper gives a differentially private algorithm for learning axis-aligned rectangles with a sample complexity that is linear in the underlying dimension improving the bound by a factor of \sqrt{d} on prior works. This is accomplished via a neat trick that might find further applications in the design of private learning algorithms. The reviewers thought that the writing was rushed and at times sloppy but the contributions were interesting and substantial. We recommend acceptance.